# Enzyme-Treated Soybean Meal Enhanced Performance via Improving Immune Response, Intestinal Morphology and Barrier Function of Nursery Pigs in Antibiotic Free Diets

**DOI:** 10.3390/ani11092600

**Published:** 2021-09-04

**Authors:** Shenfei Long, Jiayu Ma, Xiangshu Piao, Yuxin Li, Simone Husballe Rasmussen, Li Liu

**Affiliations:** 1State Key Laboratory of Animal Nutrition, College of Animal Science and Technology, China Agricultural University, Beijing 100193, China; longshenfei@cau.edu.cn (S.L.); jiayuma@cau.edu.cn (J.M.); 2Hamlet Protein A/S, Saturnvej 51, DK-8700 Horsens, Denmark; eli@HamletProtein.com (Y.L.); shr@hamletprotein.com (S.H.R.); 3Tianjin Zhongsheng Feed Co. Ltd., Tianjin 300380, China; tjzssl@126.com

**Keywords:** antioxidant status, enzyme-treated soybean meal, immune response, intestinal barrier function, performance

## Abstract

**Simple Summary:**

Currently, although extruded full-fat soybean (EFS) and enzyme-treated soybean meal (ESBM) are both commonly used plant proteins in the diets of nursery pigs, there are few studies focusing on comparing the effect of ESBM and EFS on immune response and gut development of pigs. This study found that ESBM replacing EFS could enhance performance by improving immune response, antioxidant status, gut morphology, and barrier function of nursery pigs in antibiotic free diets. The results revealed that ESBM could be an effective plant protein resource to alleviate weaning stress in pigs.

**Abstract:**

This study aims to investigate the effects of ESBM on performance, antioxidant status, immune response, and intestinal barrier function of nursery pigs in antibiotic free diets compared with EFS. A total of 32 Duroc × (Landrace × Yorkshire) barrows (initial body weight of 8.05 ± 0.66 kg, weaned on d 28) were selected and allocated to two treatments with 16 replicates per treatment and one pig per replicate using a complete random design. The treatments included an EFS group (basal diet + 24% EFS; EFS) and an ESBM group (basal diet + 15% ESBM; ESBM). Corn was used to balance energy and diets were iso-nitrogenous at about 18% crude protein. The experiment lasted for 14 days and pigs were slaughtered for sampling on d 14. Compared with EFS, pigs fed ESBM showed enhanced (*p* < 0.05) gain to feed ratio and average daily gain and a reduced (*p* < 0.05) diarrhea score. These pigs had increased (*p* < 0.05) contents of glutathione peroxidase, catalase, superoxide dismutase, IgG, interleukin-10, and ferric reducing ability of plasma, as well as decreased (*p* < 0.05) malondialdehyde, IL-6, IL-1β, tumor necrosis factor (TNF-α), interferon-γ, thiobarbituric acid-reactive substances, and diamine oxidase level in serum and TNF-α level in the jejunal mucosa. Moreover, these pigs also showed enhanced (*p* < 0.05) villus height/crypt depth in ileum, villus height in duodenum, protein expression of *zonula-occludens-1* in jejunal mucosa, and fecal total volatile fatty acids and butyric acid contents. In conclusion, ESBM replacing EFS could enhance performance via improving immune response, antioxidant status, gut morphology, and barrier function of nursery pigs in antibiotic free diets.

## 1. Introduction

After weaning, pigs are often faced with severe diarrhea and reduced performance, which might be due to the diets change from sow milk to plant-based solid diets and their immature digestive system [1]. Soybean meal (SBM) is a commonly used vegetable protein raw material in diets of swine. However, it contains soy antigens and many other anti-nutritional factors, which are detrimental to the health status and performance in pigs after weaning as well as the human nutrition as regards its undeclared presence in food [2,3]. After the soybean is extruded, extruded full-fat soybean (EFS) can not only supply significant amounts of both energy and protein to compound pig feeds, but also reduce the activities of urease and trypsin inhibitor (TI), which can increase the digestion, absorption, and utilization of nutrients [4]. Piao et al. [5] reported the EFS can be able to replace up to 50% of soybean meal in diet of nursery pigs.

After biological treatment, the level of anti-nutritional factors in SBM could also be reduced, which has no harmful effect on performance of piglets [6,7]. Cervantes-Pahm and Stein [8] found that enzyme-treated SBM (ESBM) has higher crude protein (CP) and amino acid content, and lower anti-nutritional factor content in comparison with SBM. In addition to reducing the content of most anti-nutritional factors, ESBM can also increase the content of small peptides, which has great potential for improving the health of nursery pigs [9]. Based on previous studies from our lab, ESBM can replace SBM or fermented SBM via enhancing performance, nutrient digestibility, antioxidant capacity, immunity, and intestinal morphology of nursery pigs [10]. Moreover, ESBM can also be used to replace antibiotics for reducing diarrhea and improving performance in nursery pigs based on the beneficial effects on antioxidant capacity, immunity, and intestinal barrier function [11].

At present, although EFS and ESBM are both popular for use in the diet of nursery pigs, there are few studies focusing on comparing the effect of ESBM and EFS on immune response and gut morphology in nursery pigs in antibiotics free diets. Therefore, the objective of this study was to determine the effect of ESBM replacing EFS on performance, immune response, antioxidant status, gut morphology, and barrier function of nursery pigs in antibiotics free diets.

## 2. Materials and Methods

The animal procedures performed in this trial were agreed with by the Institutional Animal Care and Use Committee of China Agricultural University (Beijing, China) (No. AW42601202-1-1).

### 2.1. ESBM and EFS Products

The EFS was provided by Beijing Tongli Xingke of Agricultural Science and Technology Company Limited (Beijing, China). The ESBM (HP300) was supplied by Hamlet Protein A/S (Horsens, Denmark), which was produced from enzyme treated defatted and dehulled SBM using a patented bioconversion process [10]. The protease and non-starch polysaccharides enzymes were used to produce ESBM. The anti-nutrient factors in EFS included 10.2 mg/g glycinin, 7.6 mg/g β-conglycinin, and 0.9 TIU/mg TI, while the anti-nutrient factors in ESBM included 0.30 mg/g glycinin, 0.15 mg/g β-conglycinin, and 0.75 TIU/mg TI. The EFS included 35.5% CP, 2.2% lysine, 0.54% methionine, 1.35% threonine, 0.45% tryptophan, and 1.69% valine, while the ESBM included 53.04% CP, 3.25% lysine, 0.79% methionine, 2.32% threonine, 0.73% tryptophan, and 2.54% valine.

### 2.2. Animals, Diets and Trial Design

A total of 32 Duroc × (Landrace × Yorkshire) barrows (initial body weight of 8.05 ± 0.66 kg, weaned on d 28) were selected and allocated to 2 treatments with 16 replicates per treatment, 1 pig per replicate using a complete random design. The treatments included an EFS group (basal diet + 24% EFS; EFS) and an ESBM group (basal diet + 15% ESBM; ESBM). Corn was used to balance energy and diets were iso-nitrogenous at about 18% CP. The amino acid levels in both diets are same. The nutrient levels in basal diets met the nutrient requirements recommended by NRC [12] in pigs (Table 1). The barrows had free access to water and were fed in mash form *ad libitum*.

The pigs were individually housed in metabolic cages (1.2 × 0.4 × 0.5 m^3^) with plastic slatted flooring for 14 days in FengNing Research Unit of China Agricultural University (Hebei, China). The average temperature and humidity remained at 24–28 °C and 60–70%, and the ammonium and CO_2_ concentration in house was controlled automatically. Pigs and feed were weighed on d 0 and 14 for the measurement of average daily feed intake (ADFI), average daily gain (ADG), and G:F (ADG/ADFI). The fecal score was measured at 09:00 and 16:00 h every day following the method by Pan et al. [13] as follows: 5 = feces were watery; 4 = feces were semiliquid and loose; 3 = feces were partially formed and soft; 2 = feces were slightly soft; and 1 = feces were hard.

### 2.3. Sample Collection and Processing

On d 14, vacutainer tubes (Greiner Bio-One GmbH, Kremsmunster, Austria) were used to collected the blood samples from the jugular vein (approximately 10 mL). The blood samples were centrifuged for 15 min at 4 °C and 3000× *g*. The supernatant was collected and stored at −20 °C. for analysis. Half of the pigs near the average BW each treatment were humanly slaughtered by exsanguination after electric corona. About 5 cm fragment in the middle of aseptic small intestine (from the proximal 1/3 of duodenum, the mid 1/3 of jejunum, and distal 1/3 of ileum) samples for the measurement of gut morphology. The scraped jejunal mucosa samples and fresh fecal samples were put in liquid nitrogen and stored at −80 °C rapidly.

### 2.4. Chemical Analysis

The ingredients and diets were ground via a 1-mm screen and analyzed for CP (methods 990.03), ash (methods 942.15), dry matter (DM; method 934.01), and ether extract (EE, method 920.39) according to the methods in AOAC [14]. An automatic adiabatic oxygen bomb calorimeter (Parr 1281, Automatic Energy Analyzer; Moline, IL, USA) was used to measure the Gross energy (GE). According to Van Soest et al. [15], the acid detergent fiber (ADF) and neutral detergent fiber (NDF) were measured. The amino acids except methionine and tryptophan using ion-exchange chromatography was measured by an automatic amino acid analyzer (L-8900, Automatic Amino Acid Analyzer; Hitachi, Tokyo, Japan). Tryptophan was determined using high performance liquid chromatography (Agilent1200 Series; Aligent, Santa Clara, CA, USA) after hydrolyzing with 4 N LiOH at 110 °C for 22 h. The methionine as methionine sulphone after peroxidation with performic acid and pre-column derivation using phenylisothiocyanate. According to You et al. [16] and Ma et al. [17], the glycinin, β-conglycinin, and TI concentration in EFS and ESBM were analyzed using ELISA kits.

### 2.5. Measurement of Serum Parameters and Interleukin Cytokines in Jejunal Mucosa

According to the manufacturer’s instructions (Nanjing Jiancheng Bioengineering Institute, Nanjing, Jiangsu, China), the levels of malondialdehyde (MDA), total superoxide dismutase (SOD), glutathione peroxidase (GSH-Px), and catalase (CAT) in serum were determined using assay kits. The PBS (containing protease inhibitor) was used to homogenize the jejunal mucosa samples. These samples were centrifuged for 15 min at 4 °C and 15,000× *g*, and then the supernatant was collected for analysis. Accordance with the manufacturer’s instructions (R&D Systems, Minneapolis, USA), the immunoglobulins (IgA, IgG, IgM), interleukin-1β (IL-1β), IL-6, IL-10, tumor necrosis factor (TNF-α), interferon-γ (IFN-γ), advance oxidation protein products (AOPP), thiobarbituric acid-reactive substances (TBARS), ferric reducing ability of plasma (FRAP), D-lactic acid (D-LA), and diamine oxidase (DAO) in serum, and the IL-1β, TNF-α, and IFN-γ in supernatant were measured using ELISA kits.

### 2.6. Gut Morphology

The gut morphology measurement was conducted according to the procedure of Long et al. [18]. A 10% neutral formalin buffer was used to fix the small intestinal samples for about 2 days. Then, the samples were cleaned, excised, dehydrated, and embedded in paraffin wax. Five tissue sections were placed on glass slides and stained with eosin and hematoxylin. About 20 villi and their adjoining crypts in each slice were measured with a calibrated 10-fold eyepiece divider. The villus height, crypt depth and their ratio were calculated.

### 2.7. Western Blot Analysis in Jujunal Mucosa Protein

The protein in jejunal mucosa samples was extracted by a ProteoJET Total Protein Extraction Kit (Fermentas, Hanover, MD, USA), while the protein content was measured by a Bicinchoninic Acid Protein Assay Kit (Applygen Technologies, Rockford, IL, USA). The protein extracts were separated by ten percent sodium dodecyl sulfate polyacrylamide gel electrophoresis, and moved to polyvinylidene fluoride membranes (Bio-Rad Laboratories, Hercules, CA, USA). These membranes were blocked for 1 h; washed with TBST for 3 times; incubated overnight with diluted antibodies against β-actin (Abcam, Cambridge, MA, USA), *zonula-occludens-1* (*ZO-1*), and *occludin* (Santa Cruz Biotechnology, Santa Cruz, CA, USA); and then incubated with goat anti-rabbit lgG secondary antibody for 1 h. The Odyssey infrared imaging system (LI-COR Biosciences, Lincoln, NE, USA) was used to display the target band, while the Quantity One software (Biorad Laboratories, Hercules, CA, USA) was used to perform the blot analysis. The *occludin*, *ZO-1*, and *β-actin* band intensities was measured and the occluding to *β-actin* and *ZO-1* to *β-actin* ratios were calculated.

### 2.8. The Volatile Fatty Acids (VFA) Contents in Feces

The VFA contents in fecal samples were measured by a Hewlett Packard 5890 gas chromatograph (Agilent Technologies Inc., USA). The fresh fecal samples (about 1.5 g) and sterile water (1.5 mL) were mixed and then centrifuged at 15,000× *g* and 4 °C for 15 min to get the supernatant. Then, the supernatant was transferred by a gas chromatograph sample bottle, and mixed with 200 μL meta-phosphoric acid. These samples were placed in ice for 30 min and then centrifuged at 15,000× *g* and 4 °C for 15 min.

### 2.9. Statistical Analysis

The GLM procedures of SAS (version 9.2; SAS Inst. Inc., Cary, NC, USA) [19] were used to analyzed all data following unpaired *Student-t* test. The individual pig was treated as the experimental unit. The means of treatments were calculated using the LSMEANS statement, and statistically significant differences were declared at *p* ≤ 0.05, while a tendency for significance was designated at 0.05 < *p* ≤ 0.10.

## 3. Results

### 3.1. Performance and Diarrhea Score

According to Table 2, pigs supplemented with ESBM showed enhanced (*p* < 0.05) ADG and G:F as well as reduced (*p* < 0.05) diarrhea score in comparison with pigs fed EFS.

### 3.2. Interleukin Cytokines in Jejunum Mucosa and Serum

According to Table 3, pigs supplemented with ESBM had lower (*p* < 0.05) content of TNF-α in jejunal mucosa, increased (*p* < 0.05) content of IL-10, and decreased (*p* < 0.05) contents of IL-1β, IL-6, IFN-γ, and TNF-α in serum in comparison with pigs fed EFS.

### 3.3. Antioxidant Status and Immunoglobulins in Serum

According to Table 4, pigs supplemented with ESBM showed increased (*p* < 0.05) contents of IgG, SOD, GSH-Px, CAT, and decreased (*p* < 0.05) MDA level in serum in comparison with pigs fed EFS.

### 3.4. Biochemical Index in Serum

According to Table 5, pigs supplemented with ESBM had increased (*p* < 0.05) content of FRAP, and decreased (*p* < 0.05) contents of TBARS and DAO in serum in comparison with pigs fed EFS.

### 3.5. Intestinal Morphology

According to Table 6, pigs fed ESBM had increased (*p* < 0.05) villus height in duodenum and villus height/crypt depth in ileum, as well as decreased (*p* < 0.05) crypt depth in ileum in comparison with pigs fed EFS. Pigs fed ESBM tended to increase (*p* = 0.08) villus height and villus height/crypt depth in jejunum in comparison with pigs fed EFS.

### 3.6. ZO-1 and Occludin Protein Expression in Jejunal Mucosa

According to Table 7 and Figure 1, pigs fed ESBM showed enhanced (*p* < 0.05) the protein expression of *ZO-1* in jejunal mucosa compared with pigs fed EFS.

### 3.7. The VFA Contents in Feces

According to Table 8, pigs supplemented with ESBM showed increased (*p* < 0.05) contents of butyric acid and total VFA in feces compared with pigs fed EFS.

## 4. Discussion

The SBM is an important part of feed formulation for farm animals. However, since SBM contains anti-nutritional factors (TI and indigestible proteins such as glycine and β-conglycine) and antigenic proteins, nursery pigs are sensitive to SBM, resulting in reduced nutrient digestion and absorption, increased diarrhea rate and thus decreased performance [2]. Fermentation and enzymatic pre-processing can reduce TI and indigestible proteins, improve the contents of amino acids of SBM, thus enhance growth performance and feed utilization [8,20,21]. Besides, the ESBM (HP300) contain less sugars and more small bioactive peptides compared with other plant proteins, as well as greater content of small peptide and less content of raffinose, stachyose and TI than SBM [10]. The EFS are demonstrated effectively in increasing the digestion, absorption and utilization of nutrients [4], while ESBM could replace SBM or fermented SBM via enhancing performance, nutrient utilization and gut morphology of nursery pigs [10]. According to the analysis, we found that ESBM has higher concentrations of CP, essential amino acid and non-essential amino acid, while ESM has lower levels of EE, NDF, ADF, and GE than EFS. While the concentrations of TI, β-conglycinin, and glycinin were lower in ESBM compared with EFS. Between both of plant proteins, the current study found pigs fed ESBM showed increased ADG and G:F, as well as decreased diarrhea score compared with pigs fed EFS, which indicated the effect of ESBM was better than EFS in alleviating weaning stress in nursery pigs. This finding might be due to the concentrations of glycinin, β-conglycinin, and TI were lower in ESBM compared with EFS. Navarro et al. [22] reported that heat processing in producing EFS might reduce the digestibility of nutrients since the high level of TI might reduce the apparent ileal digestibility of amino acids in pigs [23]. The ESBM had been demonstrated effectively in improving ADG in nursery pigs [24,25]. Moreover, Li et al. [26] had also reported that replacing half of EFS with ESBM in diets could increase the dietary nitrogen digestibility, preventing protein fermentation in the intestine, which might improve intestinal health and reduce diarrhea in pigs. Ruckman et al. [27] reported that ESBM could positively impact fecal score, while a previous study in our lab also demonstrated that ESBM could replace antibiotics in reducing diarrhea and improving performance in nursery pigs [11], which explained the current findings in antibiotic free diets.

After weaning, the pigs might face severe oxidative stress, which might be due to the over production of reactive oxidative species that could lead to redox imbalance, lipid peroxidation, protein synthesis, and DNA replication disorder. The TBARS and MDA are the production of lipid peroxidation, which reflect the oxidative status in pigs. The FRAP reflected the redox balance in nursery pigs. The SOD, GSH-Px, and CAT are anti-oxidative enzymes, which could directly alleviate peroxidation in lipid, protein, and DNA. Zhou et al. [9] reported that 1.5% ESBM could enhance the antioxidant capacity. Previous studies also point out that ESBM can replace SBM, fermented SBM, or fish meal in nursery pig diets by improving antioxidant capacity of nursery pigs [10,27]. In the current study, we reported that pigs supplemented with ESBM had enhanced FRAP, SOD, GSH-Px, CAT, and decreased TBARS and MDA levels in serum compared with EFS, which reflected the anti-oxidative capacity in ESBM was better than EFS. The reason might also be due to the lower concentrations of glycinin and β-conglycinin and TI were lower in ESBM.

For nursery pigs, high level of indigestible proteins in diet might result in inflammatory response, especially increasing pro-inflammatory cytokines levels (such as TNF-α, IL-1β and IL-6), which might decrease the gut integrity [28]. In this study, pigs supplemented with ESBM had lower IL-1β, IL-6, and IFN-γ levels in serum, suggesting that the effect of ESBM on reducing pro-inflammatory cytokines was better than EFS. The IL-10 is one of the anti-inflammatory cytokines, which could prevent damage and infection in pigs [29]. In this study, pigs supplemented with ESBM had increased content of IgG and IL-10 in serum, which suggested the enhancement of the humoral immune response by ESBM. One of the possible reasons for this finding might be ESBM can accelerate the development level of related organs and activate cells, so as to improve the immune level in pigs [9]. Another reason for the current finding might be that the ESBM had lower glycinin, β-conglycinin, and TI than EFS, which might help improve the concentration of mucosal interleukin-4 [27] and modulate the microbiota composition in nursery pigs [26].

The gut barrier integrity plays an important role in reducing negative impact of antigens and pathogens into the mucosa [30]. The serum DAO is a reliable marker of intestinal barrier integrity [31]. In the current study, we also found pigs fed ESBM had decreased content of DAO in serum, which indicated ESBM could help improve intestinal barrier integrity. Normally, tight junction barrier contains protein complexes, which could be modulated by cytokines [32,33]. Ma et al. [10] and Ruckman et al. [27] found pigs fed ESBM showed increased *ZO-1* and *occludin* protein expression in jejunal mucosa in comparison with pigs fed SBM. In the current study, we also found that pigs fed ESBM showed increased protein expression of *ZO-1* and decreased TNF-α level in jejunal mucosa compared with EFS. Al-Sadi et al. [28] reported pro-inflammatory cytokine in jejunal mucosa, such as TNF-α, could increase gut epithelial permeability, while the anti-inflammatory cytokine, such as IL-10, could effectively reduce the negative impact of antigens and pathogens on the gut epithelial permeability. Thus, the reduced TNF-α and IL-6 contents might relate to the improved gut barrier function in pigs.

Li et al. [2] has pointed out that hypersensitivity to dietary antigens could lead to gut morphological changes, while antigens from plant proteins might reduce villous height in nursery pigs [34]. The nutrient utilization was often happened in small intestine, therefore, the development of gut morphology is important for the nutrient digestibility [35,36]. In the current study, we found pigs fed ESBM had increased villus height/crypt depth in ileum, villus height in duodenum, and decreased crypt depth in ileum, these pigs also had increased villus height and villus height/crypt depth in jejunum, which reflected the intestinal morphology was improved by ESBM compared with EFS. Ruckman et al. [27] also reported that feeding ESBM could improve gut morphology. The reason for the current finding might be that the ESBM could improve the CP absorption and microbial protein synthesis efficiency [37], which might be beneficial for gut morphology.

The VFA, such as butyric acid, was the microbiota metabolites, which plays an important role on immunity, metabolism and intestinal barrier integrity in host [38,39]. This study reported that the pigs from ESBM had greater butyric acid and total VFA concentrations than those from EFS. The butyric acid could improve intestinal integrity via inhibiting inflammatory mediator production [40] and promote gut epithelial cell proliferation in pigs [41,42]. Ruckman et al. [27] also had similar findings, who reported that feeding ESBM could increase ileal VFA, modulate immune response and intestinal barrier function. Li et al. [26] also found that ESBM replacing half of EFS in diets could also increase total VFA content in cecal digesta. The reason for the current finding might be that ESBM could reduce TI in SBM and increase the content of small peptides, which could easily be absorbed [10,43], as well as improve the balance of gut microbiota community via preventing the abundance of pathogenic bacteria, and promoting the abundance of beneficial bacteria, such as *Lactobacillus* [26,44,45].

## 5. Conclusions

In conclusion, ESBM replacing EFS could enhance performance via improving antioxidant status, immune response, intestinal morphology, and barrier function of nursery pigs in antibiotic free diets.

## Figures and Tables

**Figure 1 animals-11-02600-f001:**
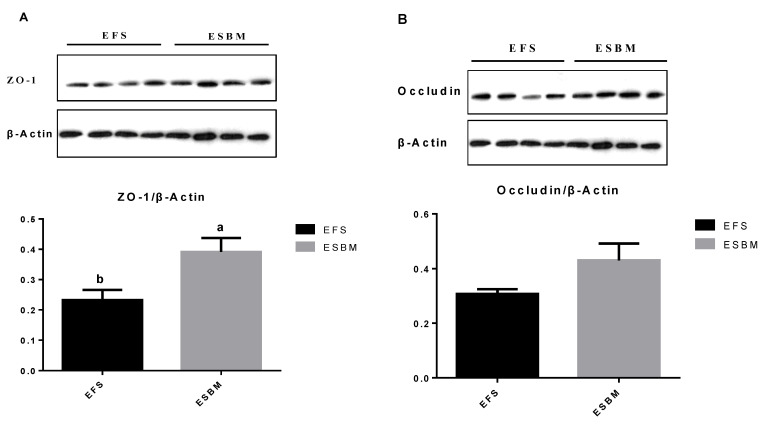
Effects of enzyme-treated soybean meal replacing extruded full-fat soybean on protein expression of *ZO-1* (**A**) and *occludin* (**B**) in jejunal mucosa in nursery pigs. *n* = 8. EFS: Extruded full-fat soybean; ESBM: Enzyme-treated SBM. ^a,b^ Different letters indicate a significant difference (*p* < 0.05). (original Western blot figures are in Appendix A.)

**Table 1 animals-11-02600-t001:** Composition and nutrient levels of the experimental diets (%, as-fed basis).

Ingredients	EFS ^1^	ESBM ^1^
Corn	32.48	41.23
Wheat	20.00	20.00
Enzyme-treated soybean meal	0.00	14.00
Extruded full-fat soybean	24.00	0.00
Whey powder	12.00	12.00
Fish meal	5.00	5.00
Soy oil	3.01	4.24
Dicalcium phosphate	1.05	0.82
Limestone	0.48	0.74
Salt	0.30	0.30
L-Lysine	0.52	0.58
Methionine	0.09	0.10
Threonine	0.18	0.17
Tryptophan	0.04	0.04
Valine	0.15	0.08
Zinc oxide	0.20	0.20
Vitamin-mineral premix ^2^	0.50	0.50
Analyzed nutrient levels		
Gross energy, MJ/kg	17.16	17.15
Crude protein	18.54	18.52
Ether extract	6.00	4.30
Neutral detergent fiber	15.50	15.00
Acid detergent fiber	2.10	3.00
Dry matter	90.00	90.50
Ash	6.40	5.40

Note: ^1^ EFS: Extruded full-fat soybean; ESBM: Enzyme-treated SBM. The anti-nutrient factors in EFS included 10.2 mg/g glycinin, 7.6 mg/g β-conglycinin and 0.9 TIU/mg trypsin inhibitor, while the anti-nutrient factors in ESBM included 0.30 mg/g glycinin, 0.15 mg/g β-conglycinin and 0.75 TIU/mg trypsin inhibitor. The EFS included 35.5% crude protein, 2.2% lysine, 0.54% methionine, 1.35% threonine, 0.45% tryptophan and 1.69% valine, while the ESBM included 53.04% crude protein, 3.25% lysine, 0.79% methionine, 2.32% threonine, 0.73% tryptophan and 2.54% valine. ^2^ Vitamin-mineral premix for each kg diet: vitamin A, 12,000 IU; vitamin E, 30 IU; vitamin D_3_, 2500 IU; vitamin K_3_, 30 mg; vitamin B_12_, 12 μg; riboflavin, 4 mg; pantothenic acid, 15 mg; nicotinic acid, 40 mg; choline chloride, 400 mg; folic acid, 0.7 mg; vitamin B_1_, 1.5 mg; vitamin B_6_, 3 mg; biotin, 0.1 mg; manganese, 40 mg; iron, 90 mg; zinc, 100 mg; copper, 8.8 mg; iodine, 0.35 mg; selenium, 0.3 mg.

**Table 2 animals-11-02600-t002:** Effects of enzyme-treated soybean meal replacing extruded full-fat soybean on performance of nursery pigs.

Items	EFS ^1^	ESBM ^1^	SEM	*p*-Value
Average daily gain, g	366.16	445.18	10.26	0.01
Average daily feed intake, g	562.75	550.42	16.07	0.40
Gain to feed ratio, g/g	0.65	0.81	0.02	<0.01
Diarrhea score	3.22	2.46	0.04	<0.01

Note: SEM means standard error of the mean. *n* = 16. ^1^ EFS: Extruded full-fat soybean; ESBM: Enzyme-treated SBM.

**Table 3 animals-11-02600-t003:** Effects of enzyme-treated soybean meal replacing extruded full-fat soybean on serum antioxidant status and immunoglobulins in nursery pigs.

Items	EFS ^1^	ESBM ^1^	SEM	*p*-Value
Antioxidant status				
Superoxide dismutase (U/mL)	125.01	136.46	2.72	0.01
Glutathione peroxidase (U/mL)	275.33	319.00	11.38	0.02
Catalase (U/mL)	1.30	2.65	0.32	0.03
Malondialdehyde (nmol/mL)	4.63	3.72	0.20	0.01
Immunoglobulins				
Immunoglobulin A (g/L)	0.91	1.01	0.09	0.55
Immunoglobulin M (g/L)	0.90	1.09	0.08	0.16
Immunoglobulin G (g/L)	6.03	7.33	0.48	0.05

Note: SEM means standard error of the mean. *n* = 8. ^1^ EFS: Extruded full-fat soybean; ESBM: Enzyme-treated SBM.

**Table 4 animals-11-02600-t004:** Effects of enzyme-treated soybean meal replacing extruded full-fat soybean on interleukin cytokines in serum and jejunal mucosa of nursery pigs (pg/mg).

Items	EFS^1^	ESBM ^1^	SEM	*p*-Value
Serum				
Interukin-1β	20.32	17.97	0.41	<0.01
Interukin-6	131.06	111.85	5.10	0.05
Interukin-10	21.68	27.37	1.54	0.02
Interferon-γ	79.82	62.97	4.00	0.02
Tumor nuclear factor-α	143.76	117.59	5.58	<0.01
Jejunal mucosa				
Interukin-1β	8.98	7.02	0.73	0.13
Interukin-6	63.20	56.57	3.13	0.21
Interukin-10	18.57	18.11	1.06	0.77
Interferon-γ	34.29	28.61	2.03	0.12
Tumor nuclear factor-α	57.98	39.83	4.62	0.05

Note: SEM means standard error of the mean. *n* = 8. ^1^ EFS: Extruded full-fat soybean; ESBM: Enzyme-treated SBM.

**Table 5 animals-11-02600-t005:** Effects of enzyme-treated soybean meal replacing extruded full-fat soybean on serum biochemical indicators in nursery pigs.

Items ^1^	EFS ^2^	ESBM ^2^	SEM	*p*-Value
AOPP (pmol/L)	85.93	81.45	1.76	0.47
TBARS (ng/mL)	56.38	42.51	1.20	<0.01
FRAP (mmol/L)	0.20	0.26	0.01	<0.01
D-LA (μM)	2.12	1.83	0.19	0.30
DAO (U/mL)	5.12	3.75	0.26	<0.01

Note: SEM means standard error of the mean. ^1^ AOPP: advance oxidation protein products; TBARS: thiobarbituric acid-reactive substances; FRAP: ferric reducing ability of plasma; D-LA: D-lactic acid; DAO: diamine oxidase. *n* = 8. ^2^ EFS: Extruded full-fat soybean; ESBM: Enzyme-treated SBM.

**Table 6 animals-11-02600-t006:** Effects of enzyme-treated soybean meal replacing extruded full-fat soybean on intestinal morphology in nursery pigs.

Items	EFS ^1^	ESBM ^1^	SEM	*p*-Value
Duodenum				
Villus height, μm	281.42	316.36	3.61	<0.01
Crypt depth, μm	180.40	192.84	10.36	0.42
Villus height/crypt depth	1.57	1.69	0.10	0.42
Jejunum				
Villus height, μm	269.38	318.30	15.43	0.08
Crypt depth, μm	157.45	145.11	8.46	0.35
Villus height/crypt depth	1.77	2.22	0.15	0.08
Ileum				
Villus height, μm	234.97	231.55	13.53	0.86
Crypt depth, μm	127.87	108.37	3.88	<0.01
Villus height/crypt depth	1.84	2.13	0.08	0.03

Note: SEM means standard error of the mean. *n* = 8. ^1^ EFS: Extruded full-fat soybean; ESBM: Enzyme-treated SBM.

**Table 7 animals-11-02600-t007:** Effects of enzyme-treated soybean meal replacing extruded full-fat soybean on protein expression of tight junction protein in jejunal mucosa in nursery pigs.

Items	EFS ^1^	ESBM ^1^	SEM	*p*-Value
*Zonula-occludens-1*/*β-Actin*	0.23	0.39	0.04	0.04
*Occludin*/*β-Actin*	0.31	0.43	0.05	0.13

Note: SEM means standard error of the mean. *n* = 8. ^1^ EFS: Extruded full-fat soybean; ESBM: Enzyme-treated SBM.

**Table 8 animals-11-02600-t008:** Effects of enzyme-treated soybean meal replacing extruded full-fat soybean on the volatile fatty acids composition in feces (mg/g).

Items	EFS ^1^	ESBM ^1^	SEM	*p*-Value
Lactic acid	0.13	0.22	0.09	0.52
Acetic acid	4.44	5.33	0.33	0.10
Propionic acid	1.85	2.31	0.18	0.11
Formic acid	0.03	0.03	0.01	0.18
Isobutyric acid	0.14	0.18	0.02	0.13
Butyric acid	0.97	1.81	0.22	0.03
Isovaleric acid	0.08	0.11	0.01	0.13
Valeric acid	0.19	0.30	0.05	0.16
Total volatile fatty acid	7.69	10.07	0.68	0.04

Note: SEM means standard error of the mean. *n* = 8. ^1^ EFS: Extruded full-fat soybean; ESBM: Enzyme-treated SBM.

## Data Availability

All data is contained within the article.

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
