# Peer review of "Enzyme-Treated Soybean Meal Enhanced Performance via Improving Immune Response, Intestinal Morphology and Barrier Function of Nursery Pigs in Antibiotic Free Diets"

_animals, 2021, doi:10.3390/ani11092600_

Round 1

Reviewer 1 Report

1.- The inclusion of raw materials (EFS 24%) are high in piglet feed two weeks after weaning at 28 of age.
Do you consider new experiments with low levels of EFS and ESBM (20 and 10)?

2.- In both experimental diets include Zinc oxide at 2 kg/Tm without antibiotics.
Do you have any interest to evaluate de diets with zinc at nutritional levels?

3.- The level of vitamin E (30 IU) are very low in my opinion (100 ?)

4.- Only for curiosity: the origin of minerals are organic or inorganic?

5.- Line 97: The barrows were drunk and fed (in a mash form) ad libitum.
Drink vs drunk

Author Response

Revision Note, (List of modification) Date: 2021-08-07 (y-m-d)

Manuscript ID: animals-1313329-R1.

Dear Sir

Good day. Thank you very much for your kind consideration with our submitted article and offering us the further opportunity to submit the revised manuscript. Please find here the point to point expert reviewer’s and editor’s comments with necessary changes as per suggested with this attached file, and the amendments are highlighted in red in the revised manuscript. We have revised our manuscript for language and grammar checked by a native English speaker working in our University. We do thanks to skilled reviewers, academic editors and editorial board members as well for their critical evaluation to make the manuscript more effective for review process in Animals Journal.

Many thanks.

Sincerely yours,

Prof. Dr. Xiang Shu Piao,

State Key laboratory of Animal Nutrition, College of Animal Science and Technology, China Agricultural University, Beijing 100193, China

Corresponding Author,

Email: [email protected]; Tel.: +86-1062733588/Fax.: +86-1062733688

Reviewer 1

Comments: 1.- The inclusion of raw materials (EFS 24%) are high in piglet feed two weeks after weaning at 28 of age.

Do you consider new experiments with low levels of EFS and ESBM (20 and 10)?

Response: Thanks for the suggestion, we will consider new experiments with low levels of EFS and ESBM (20 and 10) in our next experiment.

Comments: 2.- In both experimental diets include Zinc oxide at 2 kg/Tm without antibiotics.

Do you have any interest to evaluate de diets with zinc at nutritional levels?

Response: Thanks for the suggestion. The level of zinc oxide was added according to the recommendation by Chinese government. We will evaluate the diets with zinc at nutritional levels in the future.

Comments: 3.- The level of vitamin E (30 IU) are very low in my opinion (100?)

Response: Thanks for the suggestion, we have checked the level of vitamin E, it is 30 IU in this study.

Comments: 4.- Only for curiosity: the origin of minerals are organic or inorganic?

Response: Thanks for the suggestion. The minerals are inorganic

Comments: 5.- Line 97: The barrows were drunk and fed (in a mash form) ad libitum.

Drink vs drunk

Response: Thanks for the suggestion, we have revised according to the advice. This sentence has been changed into “The barrows had free access to water and were fed in a mash form ad libitum.”

Reviewer 2 Report

general remarks:

The paper describes many relevant parameters in relation to weaning diarrhea. The will have more scientific value if the differences in composition of ESBM and EFS would be used to explain the differences in animal response. In the discussion the authors have limited themselves to mention some of the differences in the first paragraph in the discussion. However, other relevant differences are not mentioned and/or no values were provided  in the product description. I refer to the smaller bioactive peptides and the sugars like raffinose, stachyose and verbascose. 

I would have preferred an experiment in which a control without ZnO was included. this next to the fact that the animals were housed individually could have masked some of the differences which would be seen under more practical conditions.  

I miss in the reference the patent number in which the ESBM is described. I could not find what enzyme treatment was used or is the term enzyme treatment misleading and is it a a water extraction as  mentioned in Hamlet patent : A processed protein product, Publication number 20200054041 ?

specific remarks.

line 50: contains soy antigens

line 54 : which toxins?

line 59: the higher CP content is relevant. What is removed ?

line 91 and following: although it is not mentioned were the amino acid levels in both diets the same?

line 97 should be altered. Drunken piglets? (:

table 1. It is strange to see in current research diets without phytase. What was the reason for that ?

Table 6. What is the value of measuring VFA's in the feces. Depending on the place in the colon the VFA concentration and the production differs. Also did the authors measure the feces production?

line 342: Does butyric acid also have an benificial effect if it is formed in the colon? please give reference.

Author Response

Revision Note, (List of modification) Date: 2021-08-07 (y-m-d)

Manuscript ID: animals-1313329-R1.

Dear Sir

Good day. Thank you very much for your kind consideration with our submitted article and offering us the further opportunity to submit the revised manuscript. Please find here the point to point expert reviewer’s and editor’s comments with necessary changes as per suggested with this attached file, and the amendments are highlighted in red in the revised manuscript. We have revised our manuscript for language and grammar checked by a native English speaker working in our University. We do thanks to skilled reviewers, academic editors and editorial board members as well for their critical evaluation to make the manuscript more effective for review process in Animals Journal.

Many thanks.

Sincerely yours,

Prof. Dr. Xiang Shu Piao,

State Key laboratory of Animal Nutrition, College of Animal Science and Technology, China Agricultural University, Beijing 100193, China

Corresponding Author,

Email: [email protected]; Tel.: +86-1062733588/Fax.: +86-1062733688

Reviewer 2

Comments: general remarks:

The paper describes many relevant parameters in relation to weaning diarrhea. The will have more scientific value if the differences in composition of ESBM and EFS would be used to explain the differences in animal response. In the discussion the authors have limited themselves to mention some of the differences in the first paragraph in the discussion. However, other relevant differences are not mentioned and/or no values were provided in the product description. I refer to the smaller bioactive peptides and the sugars like raffinose, stachyose and verbascose.

I would have preferred an experiment in which a control without ZnO was included. this next to the fact that the animals were housed individually could have masked some of the differences which would be seen under more practical conditions. I miss in the reference the patent number in which the ESBM is described. I could not find what enzyme treatment was used or is the term enzyme treatment misleading and is it a water extraction as mentioned in Hamlet patent: A processed protein product, Publication number 20200054041?

Response: Thanks for the suggestion, we have revised according to the advice. We will consider a trial in which a control without ZnO was included in our next study. Moreover, the enzymes for the production of HP300 included the protease and non-starch polysaccharides (NSP) enzymes. It did not have the water extraction progress during the production. According to previous study in our group, the ESBM (HP300) contain less sugars and more small bioactive peptides compared with other plant proteins. In this study, we focus more on the difference of anti-nutrient factors (such as glycinin and β-conglycinin) and all essential amino acids contents between EFS and ESBM, which might also point out ESBM had more beneficial effect than EFS. In the material and methods, we noted the main difference between EFS and ESBM in Line 84-90. Please check.

According to previous studies in our group, Ma et al. (2019) point out that enzyme-treated soybean meal had greater content of small peptide than soybean meal and less content of raffinose, stachyose and trypsin inhibitor (TI) than soybean meal. The contents of glycinin and β-conglycinin in ESBM were less than in soybean meal. We added this part in the discussion, please refer to Line 272-275.

References:

Ma XK, Shang QH, Hu JX, Liu HS, Brøkner C, Piao XS. 2019. Effects of replacing soybean meal, soy protein concentrate, fermented soybean meal or fish meal with enzyme-treated soybean meal on growth performance, nutrient digestibility, antioxidant capacity, immunity and intestinal morphology in weaned pigs. Livest Sci. 225:39-46. doi:10.1016/j.livsci.2019.04.016

Ma XK, Shang QH, Wang QQ, Hu JX, Piao XS. 2019. Comparative effects of enzymolytic soybean meal and antibiotics in diets on growth performance, antioxidant capacity, immunity, and intestinal barrier function in weaned pigs. Anim Feed Sci Technol. 248:47-58. doi:10.1016/j.anifeedsci.2018.12.003

Comments: specific remarks. Line 50: contains soy antigens

Response: Thanks for the suggestion, we have revised according to the advice.

Comments: line 54: which toxins?

Response: Thanks for the suggestion, we have corrected this part into “but also reduce the activities of urease and trypsin inhibitor (TI)” (Line 54-55) since the toxin problem is not that common in soybean. Please check.

Comments: line 59: the higher CP content is relevant. What is removed?

Response: Thanks for the suggestion, since the enzyme was used to make to make more small peptides, and amino acid, the CP content became higher than SBM. According to the study by Ma et al. (2019), the CP content in ESBM (53%) was higher than SBM (45.2%).

Reference: Ma XK, Shang QH, Hu JX, Liu HS, Brøkner C, Piao XS. 2019. Effects of replacing soybean meal, soy protein concentrate, fermented soybean meal or fish meal with enzyme-treated soybean meal on growth performance, nutrient digestibility, antioxidant capacity, immunity and intestinal morphology in weaned pigs. Livest Sci. 225:39-46. doi:10.1016/j.livsci.2019.04.016

Comments: line 91 and following: although it is not mentioned were the amino acid levels in both diets the same?

Response: Thanks for the suggestion, we have revised according to the advice. The amino acid levels in both diets are same. We have added this part in the Materials and Methods. Please refer to Line 97.

Comments: line 97 should be altered. Drunken piglets? (:

Response: Thanks for the suggestion, we have revised according to the advice. This part has been changed as following: The barrows had free access to water and were fed in a mash form ad libitum.

Comments: table 1. It is strange to see in current research diets without phytase. What was the reason for that?

Response: Thanks for the suggestion, in this study, we did not add phytase in the basal diet since we added some inorganic phosphorus to meet the phosphorus requirement by NRC 2012. Thanks for your kind advice, we will consider adding phytase in our next trial.

Comments: Table 6. What is the value of measuring VFA's in the feces. Depending on the place in the colon the VFA concentration and the production differs. Also did the authors measure the feces production?

Response: Thanks for the suggestion, we did not measure the feces production, we mainly measured the VFA concentration in the feces (Table 8) to reflect the VFA levels in large intestine and try to explain the production and utilization of VFA in large intestine for weaned pigs.

Comments: line 342: Does butyric acid also have a beneficial effect if it is formed in the colon? please give reference.

Response: Thanks for the suggestion, previous studies demonstrated that butyric acid also have a beneficial effect in large intestine (including colon) for animals since it plays an important role on immunity, metabolism and intestinal barrier integrity. In this study, we mainly measured the VFA concentration in the feces (Table 8) to reflect the VFA levels in large intestine and try to explain the production and utilization of VFA in large intestine for weaned pigs.

References:

Guilloteau P, Martin L, Eeckhaut V, Ducatelle R, Zabielski R, Van Immerseel F. 2010. From the gut to the peripheral tissues: The multiple effects of butyrate. Nutr Res Rev. 23:366-384. doi:10.1017/S0954422410000247

Reviewer 3 Report

Manuscript ID: animals-1313329

Enzyme-treated Soybean Meal Enhanced Performance via Improving Immune Response, Intestinal Morphology and Barrier Function of Nursery Pigs in Antibiotic Free Diets

I think it's a very interesting and very important topic in the farming and  one-health context nowadays as regards different sources of plant proteins in the diets of nursery pigs.   

The manuscript evaluate the effects of enzyme-treated soy-bean meal (ESBM) on performance, antioxidant status, immune response, and intestinal barrier function of nursery pigs in antibiotic free diets compared with extruded full-fat soybean (EFS).

The topic is of interest for the academics and for the people because of the results obtained and its application in field. There are some studies like this in literature, but not specific in this kind of product.  The research is well performed, the sampling and analysis were well done.

Statistical analysis was well performed;

The conclusions are of interest

The manuscript is well written and easy to understand by readers. I believe that this manuscript does not need big changes but I think you can publish the manuscript after minor revision and in the  discussions if it’s possible, you must limit the abbreviations.

Specific suggestions

Line 50-52 …it had soy antigens and many other anti-nutritional factors, which are detrimental to the health status and performance in pigs after weaning.. Soy antigens may give problems also in human nutrition as regards its undeclared presence in food: please cite PICCOLO, FILOMENA, VOLLANO, LUCIA, Base, Giuseppe, GIRASOLE, MARIAGRAZIA, SMALDONE, GIORGIO, CORTESI, MARIA LUISA (2016). Research of soybean and Lactose in meat products and preparations sampled at retail. ITALIAN JOURNAL OF FOOD SAFETY, vol. 5, ISSN: 2239-7132, doi: 10.4081/ijfs.2016.5780

Line 91… body weight of 8.05 ± 0.66 kg, weaned at d 28…  ± SD? Or standard error? Please explain

Author Response

Revision Note, (List of modification) Date: 2021-08-07 (y-m-d)

Manuscript ID: animals-1313329-R1.

Dear Sir

Good day. Thank you very much for your kind consideration with our submitted article and offering us the further opportunity to submit the revised manuscript. Please find here the point to point expert reviewer’s and editor’s comments with necessary changes as per suggested with this attached file, and the amendments are highlighted in red in the revised manuscript. We have revised our manuscript for language and grammar checked by a native English speaker working in our University. We do thanks to skilled reviewers, academic editors and editorial board members as well for their critical evaluation to make the manuscript more effective for review process in Animals Journal.

Many thanks.

Sincerely yours,

Prof. Dr. Xiang Shu Piao,

State Key laboratory of Animal Nutrition, College of Animal Science and Technology, China Agricultural University, Beijing 100193, China

Corresponding Author,

Email: [email protected]; Tel.: +86-1062733588/Fax.: +86-1062733688

Reviewer 3:

Manuscript ID: animals-1313329

Enzyme-treated Soybean Meal Enhanced Performance via Improving Immune Response, Intestinal Morphology and Barrier Function of Nursery Pigs in Antibiotic Free Diets

Comments: I think it's a very interesting and very important topic in the farming and one-health context nowadays as regards different sources of plant proteins in the diets of nursery pigs.

The manuscript evaluated the effects of enzyme-treated soy-bean meal (ESBM) on performance, antioxidant status, immune response, and intestinal barrier function of nursery pigs in antibiotic free diets compared with extruded full-fat soybean (EFS).

The topic is of interest for the academics and for the people because of the results obtained and its application in field. There are some studies like this in literature, but not specific in this kind of product. The research is well performed, the sampling and analysis were well done.

Statistical analysis was well performed;

The conclusions are of interest

The manuscript is well written and easy to understand by readers. I believe that this manuscript does not need big changes but I think you can publish the manuscript after minor revision and in the discussions if it’s possible, you must limit the abbreviations.

Response: Thanks for the suggestion, we have revised according to the advice.

Comments: Specific suggestions

Line 50-52 …it had soy antigens and many other anti-nutritional factors, which are detrimental to the health status and performance in pigs after weaning. Soy antigens may give problems also in human nutrition as regards its undeclared presence in food: please cite PICCOLO, FILOMENA, VOLLANO, LUCIA, Base, Giuseppe, GIRASOLE, MARIAGRAZIA, SMALDONE, GIORGIO, CORTESI, MARIA LUISA (2016). Research of soybean and Lactose in meat products and preparations sampled at retail. ITALIAN JOURNAL OF FOOD SAFETY, vol. 5, ISSN: 2239-7132, doi: 10.4081/ijfs.2016.5780

Response: Thanks for the suggestion, we have cited this reference as following: “Piccolo F, Vollano L, Base G, Girasole M, Smaldone G, Cortesi ML. 2016. Soybean and lactose in meat products and preparations sampled at retail. Italian Journal of Food Safety, 5: 2239-7132. doi: 10.4081/ijfs.2016.5780.” Please refer to Line 52-53.

Comments: Line 91… body weight of 8.05 ± 0.66 kg, weaned at d 28… ± SD? Or standard error? Please explain

Response: Thanks for the kind reminding, in this study, it is weaned at d 28… ± standard error. Please check
